# In vivo evaluation of histopathologic findings of vascular damage after mechanical thrombectomy with the Tromba device in a canine model of cerebral infarction

**Sang-Hun Lee**[1]*, **Sang Woo Kim**[2], **Jong Min Kim**[2], **Woo Chan Son**[3]

1 Department of Neurology, Korea University Ansan Hospital, Korea University College of Medicine, Ansan, Republic of Korea, 2 Laboratory Animal Center, Osong Medical Innovation Foundation (K-BIO HEALTH), Cheonju, Republic of Korea, 3 Department of Pathology, University of Ulsan College of Medicine, Asan Medical Center, Seoul, Republic of Korea

* huny0029@naver.com

**Data Availability Statement:** All relevant data are within the paper and its Supporting information files.

## Abstract

A novel stent retriever device for in vivo mechanical thrombectomy for acute cerebral infarction has been developed. In this study, we compared the thrombus removal capacity, potential complications, and extent of vessel wall damage of this novel device with those of the Solitaire FR device by performing a histopathologic analysis using an autopsied canine model. Through this experimental evaluation, we aimed to assess the safety and efficacy of the newly developed thrombus removal device for cerebral infarction. Blood clots (autologous thrombus) were injected into 12 canines. Mechanical thrombectomy was performed in six canines using the newly developed Tromba thrombectomy device (experimental group) and in the other six canines using the Solitaire FR thrombectomy device (control group). Angiographic and histopathologic evaluations were performed 1 month after the blood vessels underwent mechanical thrombectomy. In the experimental group, the reperfusion patency was classified as "no narrowing" in five cases and "moderate narrowing (25%–50% stenosis)" in one case. In the control group, the reperfusion patency was classified as "no narrowing" in four cases, "moderate narrowing (25%–50% stenosis)" in one case, and "slight narrowing (less than 25% stenosis)" in one case. In the experimental group, intimal proliferation was observed in only two cases, endothelial loss was observed in two cases, and device-induced medial injury was observed in one case. In the control group, intimal proliferation was observed in two cases, endothelial loss was observed in one case, and thrombosis (fibrin/platelet) was observed in one case. The Tromba thrombectomy device showed no significant difference to the conventional Solitaire device in angiographic and histopathologic evaluations after thrombus removal. The stability and efficiency of the newly developed Tromba device are considered to be high and comparable to those of Solitaire.

**Funding:** Dr. Sang Hun Lee was supported by the Seoul R&D program (BT190099) funded by the Seoul Business Agency (SBA, Korea) and the Korea University Hospital Grant (K2210671)]. The funders had no role in study design, data collection and analysis, decision to publish, or preparation of the manuscript.

**Competing interests:** We have declared that no competing interests exist.

## Introduction

Mechanical thrombectomy has been receiving increasing attention in the treatment of ischemic stroke [1–5]. Several recent pivotal studies have investigated whether mechanical thrombectomy can accelerate recanalization and improve stroke prognosis using a self-expanding stent retriever such as the Solitaire FR Revascularization Device. These studies have demonstrated that mechanical thrombectomy leads to rapid blood flow recovery and improvement of functional outcome in acute ischemic stroke patients [1–6]. Several new thrombectomy devices with different theoretical mechanisms of action have been introduced in recent years. Stent-like devices, including various self-expandable stent retrievers such as the Solitaire flow restoration device and the Trevo retriever, are the most widely used devices for thrombectomy, which is performed as a representative treatment [7–10].

We previously performed a histopathologic analysis of wall injury in an autopsied canine model after mechanical thrombectomy using the Solitaire FR device, which is currently the most widely used mechanical thrombectomy device. In this study, we aimed to perform an experimental evaluation of the safety and efficacy of a newly developed thrombectomy device. The thrombus removal ability, potential complications, and degree of vessel wall damage of the novel stent retriever device developed for in vivo mechanical thrombectomy were compared with those of the Solitaire FR device by performing histopathologic analysis using an autopsied canine model.

## Methods

### Device description

The Tromba thrombectomy device was designed for the removal of occlusive thrombi in the setting of acute ischemic stroke, in vessels with a diameter ranging from 1.5 to 3.5 mm. It is made of nickel–titanium, which is a shape memory alloy material. The detailed dimensions of this novel device, which was used in the experimental group, are shown in Fig 1. The Tromba device is a stent for thrombus removal with a plate-shaped structure that is rolled in a spiral form and cut at regular intervals along the spiral direction around the body part (Fig 1B). Therefore, when an external force is applied, the rolled structure spreads out in the form of a plate. The front portion of the body part includes a thrombus removal section for receiving the thrombus, and the rear portion includes a delivery section that retrieves the body part into the catheter. This device has several types of cell patterns, and a single device can have a first cell pattern and a second cell pattern. The first cell pattern includes several first link parts arranged in an oblique line, spaced apart by a predetermined distance, and several second link parts spaced apart by a predetermined distance while connecting the adjacent first link parts.

In addition, the device is composed of a main cell part, which includes numerous main cells arranged in a helical shape, and numerous open cell parts arranged in a helical shape while facing in the longitudinal direction with respect to the main cell. The main cell is formed by two first link units and two second link units, and the line connecting the vertices in the left and right directions forms an angle of $\leq 45°$ with respect to the longitudinal direction of the stent.

### Animal care and process of in vivo study

To identify all experimental environments, we tested canines of the same age, weight, and size. Twelve 6-month-old canines (beagle dogs obtained from Marshall BioResources, North Rose, NY, USA) weighing between 6.0 and 8.0 kg were included in this study. Canine models have several advantages. The size and extent of vasospasm occurring in canines are more similar to

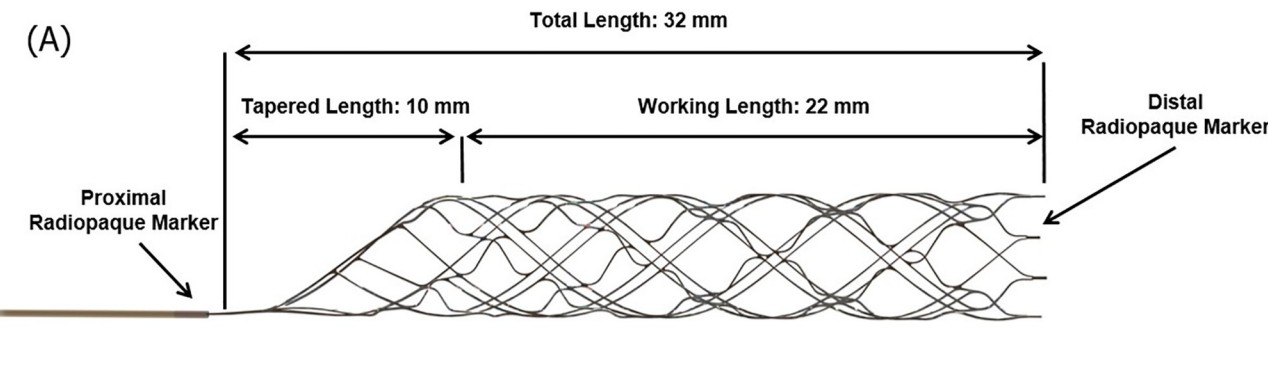

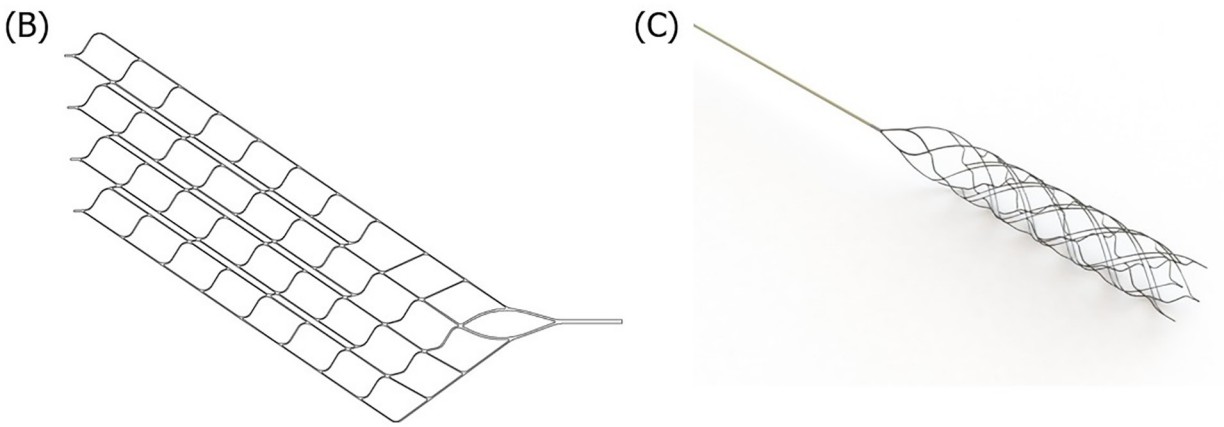

**Fig 1. Tromba device.** Schematic diagram (A), (B) and photograph (C) of the Tromba thrombectomy device.

those in humans than in porcine and rodent models, allowing for more targeted endovascular treatment and the branches of the external carotid artery in the canine model have a vessel size similar to that of the human middle cerebral artery, which is sufficient for the study of novel endovascular devices for human cerebral blood vessels [11]. One animal was bred in a cage (0.9 m$^2$) in accordance with AAALAC guidelines (0.74 m$^2$), and an individual identification card with the test and animal numbers was attached to the breeding cage. A classic dog toy was used for environmental enrichment. The conditions of the breeding environment were as follows: temperature, 23°C ± 3°C; relative humidity, 55% ± 15%; ventilation, 10–20 times/h; lighting time, 12 h (lights up at 8 am, lights out at 8 pm), and illuminance, 150–300 Lux. This study was conducted in an established canine-breeding room. For the feed supplied to the test, a solid feed for test animals (2925 Telan Global 25% Protein Dog Diet) was sterilized by irradiation. Ultrapure water produced by an RO water (reverse osmosis distilled water) production and supply device was sterilized using sodium hypochlorite and a UV device and then freely dispensed through an automatic water supply nozzle.

All experiments were conducted under general anesthesia. Preoperative anesthesia was induced by intramuscular injection of 5 mg/kg Zoletil (Bibac Korea, Seoul, Korea) and 2 mg/kg Rompun (Bayer Korea, Ansan, Korea). For postoperative pain/stress relief in experimental animals, 0.2 mg/kg of Metacam (Boehringer Ingelheim, Germany) was subcutaneously injected once daily for three days. In addition, 15 mg/kg cefazolin (Chong Kun Dang, Korea) was subcutaneously injected once daily for three days to prevent infection. For humane management of diet, exercise, and nutrition in experimental animals with stroke induced by

vascular occlusion and mechanical thrombectomy, the experiment was limited to unilateral sites of the carotid artery, not to bilateral sites. During the observation period, animals treated with the stent retriever in the breeding room of the experimental animal center were visually observed for abnormal symptoms once a day, and the experimental animals had no restrictions on their daily activities, including diet.

Histopathological analysis was performed by excising vessels that had undergone mechanical thrombectomy. Anesthesia was induced by intramuscular injection of 5 mg/kg Zoletil (Bibac Korea, Seoul, Korea) and 2 mg/kg Rompun (Bayer Korea, Ansan, Korea) in the operating room of the experimental animal center. After induction of general anesthesia, airway intubation was performed, the animal was connected to an inhalation anesthesia machine (Fabius GS premium, Drager, Germany), and respiratory anesthesia was maintained with isoflurane (<3%). The animals' anesthetic conditions (electrocardiogram, oxygen saturation, carbon dioxide concentration, etc.) were constantly monitored using a patient monitoring system (Vista 120, Drager, Germany). At the end of the experiment, 40 ml of KCl was injected intravenously after intramuscular injection to induce euthanasia, and respiration and cardiac arrest in the experimental animals were confirmed. There were no other mortalities, except planned euthanasia, and no humane endpoints in the study.

## Preparation of thrombus

The preparation and application of the thrombus followed the method applied in previous studies [12,13]. In brief, a thrombus was created by mixing 10 mL autologous blood from an animal and 25 IU bovine thrombin solution (Dade Behring, Newark, NJ, USA) in a silicone tube for 120 min at room temperature [14,15]. After refrigeration at 4˚C for 24 h, the thrombus was prepared by placing it in a 10-mm silicone tube containing saline [12,13].

## Initial and follow-up angiography, thrombus embedding, and mechanical thrombectomy protocol

We used a monoplane high-resolution angiography system (Artis Zee multi-purpose system; Siemens, Erlangen, Germany). First, a short 6-F catheter sheath (Terumo, Tokyo, Japan) was inserted into the common femoral artery, and one of the branches of the external carotid artery (occipital, superficial, or linguofacial artery) was selected. Subsequently, a pre-manufactured thrombus was injected into the target vessel using a 6-F intermediate catheter (Envoy DA; DePuy Synthes Companies, Warsaw, IN, USA) and the occlusion was maintained by removing the intermediate catheter for 15 min. After confirming the exact location of the thrombus and complete occlusion of blood vessels with angiography, thrombus removal was performed using the Tromba (4 × 20 mm; CGBio, Seongnam, Korea) thrombectomy device in the experimental group and the Solitaire FR (4 × 20 mm; EV3, Irvine, CA, USA) thrombectomy device in the control group. The degree of reperfusion was confirmed through subsequent angiography, and a follow-up examination was performed 1 month later to determine the degree of maintenance of reperfusion and the presence of vascular complications. The grade of reperfusion patency was defined according to the scale introduced by Grandin et al., as follows: no narrowing, slight narrowing (25% reduction in the lumen diameter), moderate narrowing (25%–50% stenosis or 50%–75% stenosis affecting only a short segment of the vessel), and severe narrowing (50%–75% stenosis affecting a long segment of the vessel or stenosis > 75%) [16]. All procedures were conducted according to international guidelines and were approved by the IACUC of Osong Medical Innovation Foundation in Republic of Korea (KBIO-IA-CUC-2019-051-1).

### Histopathologic evaluation

Histopathologic evaluation was performed 1 month after mechanical thrombectomy to determine the degree and condition of the recovery of wall injury using an autopsied canine model. The vessel sections from each artery were processed and embedded in paraffin, according to standard laboratory procedures. Specimens were processed for histopathologic analysis, including hematoxylin-and-eosin and Masson's trichrome staining.

To assess the degree of arterial injury, ordinal data were collected for multiple parameters, including (i) inflammation (%), (ii) intimal proliferation (%), (iii) endothelial loss (%), (iv) thrombosis (fibrin/platelet), (v) hemorrhage (adventitial/medial), and (vi) medial injury (device-induced), using the semiquantitative scoring system of Nogueira et al [17]. The degrees of inflammation, intimal proliferation, and endothelial loss were classified as "no change," "change < 25%," "change between 25% and 50%," "change between 51% and 75%," and "change > 75%." The degree of thrombosis was classified as "no change"; "minimal, focal"; "mild, multifocal"; "moderate, regionally diffuse"; and "severe, marked diffuse, or total luminal occlusion." The degree of hemorrhage was classified as "no change"; "focal, occasional"; "multifocal and regional"; "regionally diffuse"; and "100% red blood cells." Finally, the degree of medial injury was classified as "no change," "focal disruption of the internal elastic lamina," "widespread disruption of the internal elastic lamina," "medial tear not extending beyond the external elastic lamina," and "medial tear with involvement of the external elastic lamina." The sections were examined under an optical microscope at a magnification ranging from 40× to 100× by a board-certified pathologist (WC Son) who was blinded to the device used.

## Results

### Angiographic results

Using digital subtraction angiography, we selected a blood vessel with a diameter similar to that of the human middle cerebral artery as the target vessel for thrombectomy (Fig 2). Either the linguofacial artery, occipital artery, or superficial temporal artery was occluded. One blood vessel was selected from each of the 12 canines. In the experimental group (Tromba thrombectomy device), the linguofacial artery was selected in two cases, the occipital artery in three cases, and the superficial temporal artery in one case. In the control group (Solitaire FR thrombectomy device), the occipital artery was selected in five cases and the superficial temporal artery was selected in one case. In the experimental group, the reperfusion patency was classified as "no narrowing" in five cases and "moderate narrowing" in one case, and most cases were recanalized with fewer than three attempts. In the control group, the reperfusion patency was classified as "no narrowing" in four cases, "moderate narrowing" in one case, and "slight narrowing" in one case, and most cases were recanalized with fewer than two attempts. No vascular complications, such as dissection or perforation, were observed in either the experimental group or the control group (Table 1).

### Histopathologic results

Histopathologic evaluation was performed 1 month after mechanical thrombectomy. Two linguofacial arteries, eight occipital arteries, and two superficial temporal arteries with diameters of 2.2–3.1 mm from the 12 swines were evaluated. We cut the blood vessels that were subjected to thrombectomy into 4-μm sections, which were subsequently stained with hematoxylin-and-eosin and Masson's trichrome. The sections were examined under an optical microscope at 50× magnification (Fig 3).

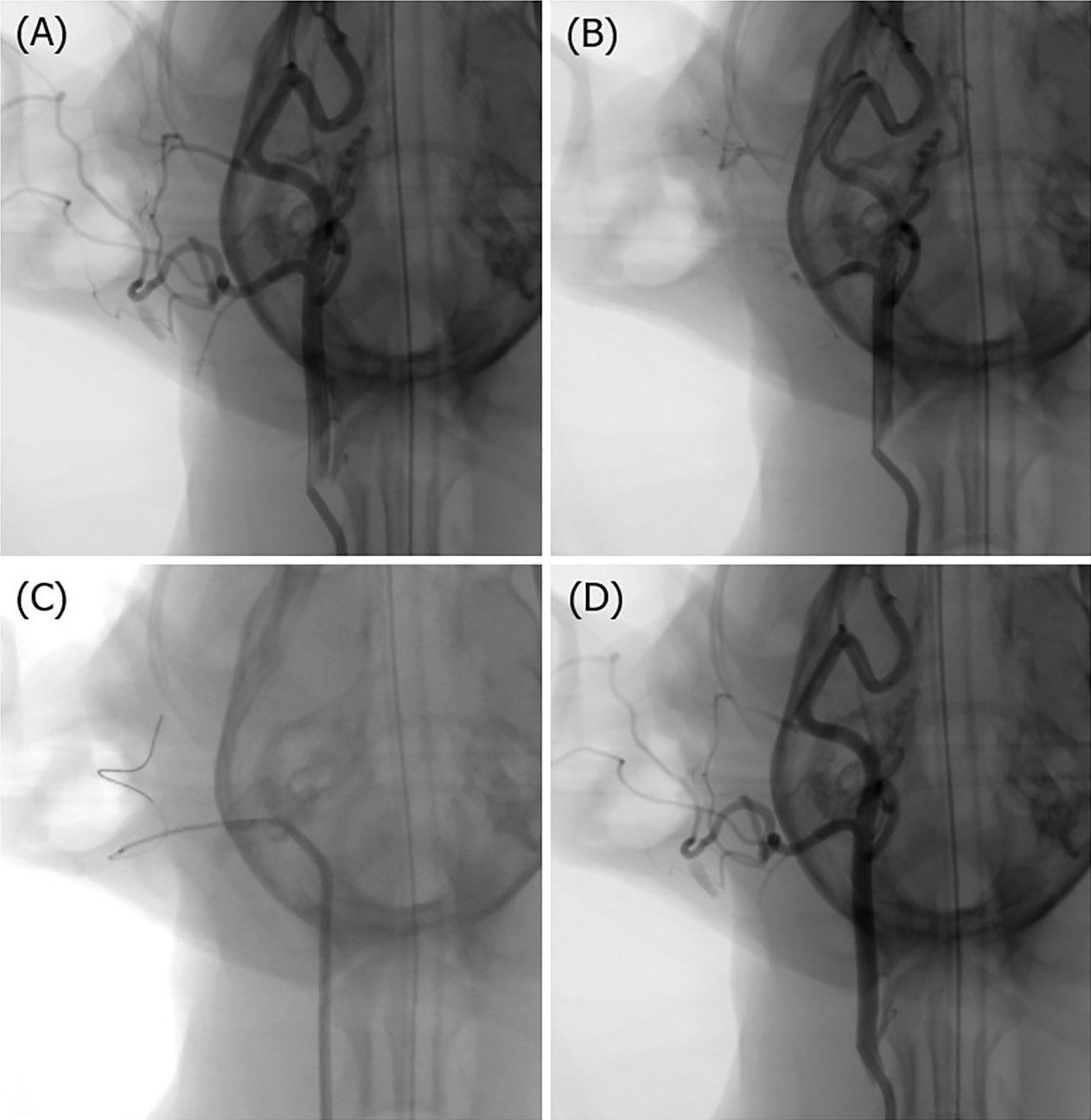

**Fig 2. Representative angiograms.** (A) Initial digital subtraction angiogram of the right common carotid artery. (B) Injection of a pre-manufactured thrombus into the target occipital artery. (C) Thrombectomy procedure. (D) Complete recanalization of the target occipital artery.

In the experimental group, inflammatory changes were not observed in any case and 25%–50% intimal proliferation was observed in only two cases (Table 2). A < 25% endothelial loss was observed in two cases, and device-induced medial injury classified as "focal disruption of the internal elastic lamina" was observed in one case. Thrombosis (fibrin/platelet) and hemorrhage (adventitial/medial) were not observed in any case in the experimental group. In the control group, serious intimal proliferation (> 75%) was observed in one case and 25%–50%

**Table 1. Demographic characteristics and presentation of six cases subjected to mechanical thrombectomy.**

| | Tromba (experimental group) | | | | | Solitaire™ FR 4×20 mm (control group) | | | |
|---|---|---|---|---|---|---|---|---|---|
| No | Occlusion site | Reperfusion patency | No. of passes / Vasc Cx | 1 M follow-up DSA | No | Occlusion site | Reperfusion patency | No. of passes / Vasc Cx | 1 M follow-up DSA |
| 1 | Rt LFA | Moderate narrowing | 3 / (-) | No interval change | 7 | Rt OA | Mo narrowing | 1 / (-) | No interval change |
| 2 | Rt LFA | No narrowing | 2 / (-) | No interval change | 8 | Rt OA | Moderate narrowing | 2 / (-) | No interval change |
| 3 | Rt OA | No narrowing | 1 / (-) | No interval change | 9 | Rt OA | No narrowing | 1 / (-) | No interval change |
| 4 | Rt OA | No narrowing | 1 / (-) | No interval change | 10 | Rt OA | No narrowing | 1 / (-) | No interval change |
| 5 | Rt OA | No narrowing | 3 / (-) | No interval change | 11 | Rt OA | Slight narrowing | 2 / (-) | No interval change |
| 6 | Rt STA | No narrowing | 1 / (-) | No interval change | 12 | Rt STA | No narrowing | 1 / (-) | No interval change |

No., number; Vasc Cx, vascular complication; M, month; DSA, digital subtraction angiography; Rt, right; LFA, linguofacial artery; OA, occipital artery; STA, superficial temporal artery. Slight narrowing, 25% reduction in the lumen diameter; moderate narrowing, 25%–50% stenosis or 50%–75% stenosis affecting only a short segment of the vessel; severe narrowing, 50%–75% stenosis affecting a long segment of the vessel or stenosis > 75%.

intimal proliferation was observed in one case. A < 25% endothelial loss was observed in one case. "Severe, marked diffuse" thrombosis (fibrin/platelet) was also observed in one case. Inflammation, hemorrhage (adventitial/medial), and device-induced medial injury were not observed in any case in the control group.

## Discussion

In this study, histologic and angiographic findings were compared between the newly developed Tromba device (experimental group) and the currently widely used Solitaire stent (control group). Tromba showed no significant difference to the conventional Solitaire device in angiographic and histopathologic findings after thrombus removal.

In general, a stent maintains the intravascular space by pushing the stenosis outward from the conduit through the radial force from its expansion in the lateral direction [18,19]. To be suitable for use in the human body, stents must have special properties in addition to expansion capacity, among which flexibility is particularly important [20]. Stent flexibility refers to the ability of a stent to be adapted to the shape of the bent portion of a blood vessel [21]. If flexibility is insufficient, the stent may lose its adaptability in the bent portion of the blood vessel, which may result in an abnormal conduit [20,21]. When a conventional flat-type stent is inserted into the catheter tube, as the rectangular thin film is inserted in a rolled state, the portion located inside the rolled shape becomes relatively severely bent and subjected to a stronger bending stress than the portion located outside. This causes the stent to return to its original shape and results in reduced wall apposition. In addition, when a conventional flat stent is deployed from the tube of the catheter and comes in close contact with the blood vessel wall, because of the difference in thickness along the circumferential direction of the rolled shape, it becomes difficult to maintain flexibility with respect to a blood vessel with several bends in various directions.

To solve the problems of the existing technology, a stent for thrombus removal with a plate-shaped structure rolled in a spiral form, consisting of a hybrid structure with open and closed cells that can provide high flexibility in blood vessels bent in various directions in three dimensions, was developed.

**Fig 3. Histopathologic evaluation.** Histopathologic evaluation was performed using two linguofacial arteries, eight occipital arteries, and two superficial temporal arteries with diameters of 2.2–3.1 mm from the 12 swines. The blood vessels that were subjected to thrombectomy were cut into 4-μm sections and stained with hematoxylin-and-eosin (H&E) and Masson's trichrome. The sections were examined under an optical microscope at 50× magnification.

The thrombus removal part of the existing thrombus retrievable stent developed so far, including Solitaire, is in the form of a closed cell structure in which all cells are connected by a link to fix and recover the thrombus. The stent of the "closed cell" structure fixes the thrombus in a relatively stable manner, so that the captured thrombus is smoothly recovered. However, due to its low flexibility and expansion power, it is difficult for this stent to adapt to the tortuous cerebral blood vessels, making it difficult to remove a thrombus in a severely bent region. This is a limitation as repeated thrombus removals can damage blood vessel walls. The hybrid cell structure is judged to be ideal because it has both the characteristics of an open cell, i.e., it can adapt well to complex and tortuous cerebral blood vessels, and a closed cell, i.e., it can stably remove and recover blood clots. In addition, a structure with a weak radial force and a

**Table 2. Semiquantitative analysis of histopathologic changes to assess the degree of arterial damage in blood vessels due to the use of Tromba and Solitaire.**

| Case no. of Tromba | Inflammation (%) | Intimal proliferation (%) | Endothelial loss (%) | Thrombosis (fibrin/platelet) | Hemorrhage (adventitial/medial) | Medial injury (device-induced) |
|---|---|---|---|---|---|---|
| 1 | None | 25% ~ 50% | <25% | None | None | None |
| 2 | None | 25% ~ 50% | None | None | None | None |
| 3 | None | None | <25% | None | None | None |
| 4 | None | None | None | None | None | None |
| 5 | None | None | None | None | None | Focal disruption of the IEL |
| 6 | None | None | None | None | None | None |
| Case no. of Solitaire | Inflammation (%) | Intimal proliferation (%) | Endothelial loss (%) | Thrombosis (fibrin/platelet) | Hemorrhage (adventitial/medial) | Medial injury (device-induced) |
| 7 | None | >75% | <25% | Severe, marked diffuse | None | None |
| 8 | None | 25% ~ 50% | None | None | None | None |
| 9 | None | None | None | None | None | None |
| 10 | None | None | None | None | None | None |
| 11 | None | None | None | None | None | None |
| 12 | None | None | None | None | None | None |

No., number; IEL, internal elastic lamina.

structure that cannot be recaptured are the biggest disadvantages of an open cell. These can be overcome with a hybrid cell structure.

The model used in this study has some limitations. First, the canine internal carotid artery is not sufficiently large to accommodate thrombectomy devices; therefore, we used external carotid artery branches, which made it difficult to apply the previously described Thrombolysis in Cerebral Infarction score for grading reperfusion. Therefore, we used the reperfusion patency grading scale suggested by Grandin et al., and we encountered no problems in comparing the reperfusion rates of the two groups. Second, as the experiment was completed with a relatively small number of canines (a total of 12 animals), the comparison of the thrombus removal stents may be insufficient and our results may be difficult to generalize. However, histologic comparisons involving a large number of animals have been conducted in other studies using swine or canine models (usually less than 10 animals) [12,15,17,22–24]. Third, imaging follow-up was performed immediately after mechanical thrombectomy and 1 month later, without regular follow-up investigations in between. Therefore, angiographic evaluation was not performed during the condition of acute phase vascular damage and blood flow. Finally, as this was a small pilot study, the results should be interpreted with caution.

In conclusion, the Tromba mechanical thrombectomy device provides high flexibility and high adhesion to the vessel wall. No significant difference was observed between the two devices compared in this study. The stability and efficiency of the newly developed Tromba device are considered to be high and comparable to those of Solitaire.

## Supporting information

**S1 Fig. Raw histopathology images.**
(PDF)

**S1 File. Experiment report.** Evaluation of safety and efficacy of stent retrievers for treatment of ischemic stroke.
(PDF)

## Author Contributions

**Conceptualization:** Sang-Hun Lee.

**Data curation:** Sang-Hun Lee, Woo Chan Son.

**Formal analysis:** Sang-Hun Lee, Woo Chan Son.

**Investigation:** Sang-Hun Lee, Sang Woo Kim, Jong Min Kim.

**Methodology:** Sang-Hun Lee, Sang Woo Kim, Jong Min Kim.

**Project administration:** Sang-Hun Lee.

**Supervision:** Sang-Hun Lee.

**Writing – original draft:** Sang-Hun Lee.

**Writing – review & editing:** Sang-Hun Lee.

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
