## [Decision Letter · Decision Letter 0]

28 Mar 2022

PONE-D-21-40394In vivo evaluation of histopathologic findings of vascular damage after mechanical thrombectomy with the Tromba device in a canine modelPLOS ONE

Dear Dr. LEE,

Thank you for submitting your manuscript to PLOS ONE. After careful consideration, we feel that it has merit but does not fully meet PLOS ONE’s publication criteria as it currently stands. Therefore, we invite you to submit a revised version of the manuscript that addresses the points raised during the review process.

We look forward to receiving your revised manuscript.

Kind regards,

Andreas Zirlik, MD

Academic Editor

PLOS ONE

Journal Requirements:

“Dr. Sang Hun Lee was supported by the Seoul R&D program (BT190099) funded by the Seoul Business Agency (SBA, Korea) and the Korea University Ansan Hospital Grant (K2110981).”

Reviewers' comments:

Reviewer's Responses to Questions

**Comments to the Author**

1. Is the manuscript technically sound, and do the data support the conclusions?

Reviewer #1: Yes

Reviewer #2: Yes

2. Has the statistical analysis been performed appropriately and rigorously? 

Reviewer #1: N/A

Reviewer #2: Yes

3. Have the authors made all data underlying the findings in their manuscript fully available?

Reviewer #1: Yes

Reviewer #2: Yes

4. Is the manuscript presented in an intelligible fashion and written in standard English?

Reviewer #1: Yes

Reviewer #2: Yes

5. Review Comments to the Author

Reviewer #1: In vivo evaluation of histopathologic findings of vascular damage after mechanical thrombectomy with the Tromba device in a canine model

Authors present in the manuscript the results of in total 12 animal experiments, where a newly developed thrombectomy device was tested (n=6) and compared with a reference device (n=6), being already in clinical use.

They found that the novel device is effective in terms of thrombus removal, an safe in terms vessel wall injury.

I have the following comments to be considered for the sake of further improvement of the manuscript.

Abstract:

In general the abstract should provide the reader with a more complete overview on the study. It should be understandable as a stand-alone piece of work, as well. Therefore the following aspects should be definitely improved:

1. It is unclear from the title, as well as from the abstract, for which vessels the device is developed. Is that a thrombectomy device for coronaries? For central nervous system? For pulmonary? For deep veins?

2. It has to be declared in the abstract, at what timepoint the animal models were sacrificed and analyzed. Considering that endothelial damage has the potential for instantaneous risk of further thrombus formation, therefore acute result is a critical aspect. On the other and it might heal over time, losing its relevance. But this would be nice to demonstrate. Since authors mention ‘intimal proliferation’, one would assume that chronic results are presented, however it is still unclear.

3. ‘Moderate-’ and ‘slight narrowing’ are quite subjective phrasing for someone who is not fully familiar with the Grandin scale and therefore difficult to interpret for comparison. Please quantify it.

Main text:

1. It would be important to define, why the Solitaire FR was selected as reference. Is that the most widely used and accepted device of the field?

2. From main text it is clear that results were ‘only’ at intermediate term after 30-days assessed. Since acute injury could have been assessed by intravascular imaging modalities, I believe the lack of acute data is a major limitation and it has to be acknowledged.

3. In the histopathologic result section the Authors write about swine model instead of canine. Please clarify.

4. In the limitation section Authors state: “However, histologic comparisons involving a large number of animals have been conducted in other canine models, and the results were considered to be of sufficient value.” It is unclear, which study and what device they refer to. It should be at least referenced or even described in a bit more details.

5. A certain structural / mechanical / design comparison of the novel versus the traditional device, focusing on potential advantages would be important for better understanding.

Main text

1. You wrore: “Lesion location was defined according to SYNTAX score”. But actually it is correct as follows: The coronary arterial segments were defined according to the American Heart Association, as modified for the ARTS I and II studies. (Ref: Serruys PW, Unger F, van Hout BA, van den Brand MJ, van Herwerden LA, van Es GA, Bonnier JJ, Simon R, Cremer J, Colombo A, Santoli C, Vandormael M, Marshall PR, Madonna O, Firth BG, Breeman A, Morel MA, Hugenholtz PG. The ARTS study (Arterial Revascularization Therapies Study). Semin Interv Cardiol 1999;4:209-19.)

2. You wrote “Kaplan-Meier (KM) cumulative survival curves were generated and the difference in survival between groups was compared by the log-rank test.”. I would suggest writing as ‘event free survival’.

3. “In the revascularization group, 105 (23%) patients were treated with CABG and 344 (77%) with PCI.” Considering, you are talking only about patients with single vessel disease, this is a relatively high rate, isn’t it.

4. “Among them, 1010 received medical therapy and 449 (30%) underwent revascularization based on the FFR measurement.” I would suggest deleting the last few words. Not all patients below 0.75 were revascularized. Not all patients below 0.80 were revascularized. Some patients above 0.80 were revascularized. Therefore saying ‘based on FFR’ is inaccurate, I believe.

5. Abstract: “Diameter stenosis, minimum lumen diameter, and FFR values were lower in rev group.” Main text: “On average, patients who received revascularization (….) had a higher percent diameter stenosis and a lower minimum lumen diameter and lower FFR as compared to patients who received medical therapy.“ Should be corrected.

6. “At a mean follow up of 3.3 years, MACE rate was similar between the medical therapy and the revascularization group.” It is unclear whether you talk about the grey zone or the entire population.

7. You wrote: “Compared to the revascularization group (Figure 2, Table 3), MACE rate was significantly higher in the 0.70-0.75 strata (p=0.047) and significantly lower in the 0.81-0.85 strata (p=0.023). There was no difference in MACE rate between the FFR gray zone strata of the medical therapy group and the revascularization group (p=0.804).”

I suggest changing it in this way: “Compared to the total revascularization group (Figure 2, Table 3), MACE rate was significantly higher in the 0.70-0.75 strata (p=0.047) and significantly lower in the 0.81-0.85 strata (p=0.023) of the MT group. There was no difference in MACE rate between the FFR gray zone strata of the medical therapy group and the revascularization group (p=0.804).”

8. Use abbreviations consequently. For instance either always MT and Rev or not at all…

9. I think the main interest why someone will read this paper to learn, what is better in the grey zone: MT or Rev. Accordingly, the structure, especially in the Results should be organized in this way.

There is lot of talks about the differences in outcome between the different strata of MT group. After Nils’ metaanalysis paper this is not much new. Therefore it is too bad that you hide the ‘million dollar answer’ in the very last sentence of the Results. It could be better the other way around, I believe.

10. “At linear regression analysis, MACE rate was overlapping between medical therapy and revascularization group at FFR value of about 0.78 (Figure 5).” It would be more accurate here to highlight the 95% CI ranges for the two lines and then you can give a range of overlap instead of an exact value.

11. If you have sufficient number of cases, it would be interesting to see what are the event rates in BMS patients vs DES patients vs CABG patients (latter most probably 99.99% LIMA-LAD, since everyone is SVD)

If type of stent is not available, then you can simply indicate it as early cases (before 2007-8?) and recent cases (after that) considering BMS era vs DES era.

12. I am not sure that Table 1 is relevant. Actually you never compare the entire MT group with the Rev group, therefore comparing clinical characteristics is unnecessary. It might be more relevant to indicate the comparison of clinical characteristics between Rev and MT strata.

13. Figure 2 – 3A – 4: where p is significant when comparing more than one groups, then one-by-one comparison should be indicated, as well.

Reviewer #2: Solid data presentation with regard to this new technology

Comments:

-With the assumption that acute stroke should be acute emboli acute thrombotic clots were injected; seems reasonable. But in reality also chronic clots could migrate to the cerebral arteries, eg patient has AF. Are there any data if this technology also works in chronic clots??

-The authors state throughout the manuscript that they used 12 canine models (beagles), but in the section histopathological findings they state that they used 12 swines?? This need to be clarified

6. PLOS authors have the option to publish the peer review history of their article (what does this mean?). If published, this will include your full peer review and any attached files.

Reviewer #1: No

Reviewer #2: No

---

## [Author Response · Author response to Decision Letter 0]

19 May 2022

Thank you for the opportunity to revise our manuscript. We appreciate your careful review and constructive proposals. We believe that the manuscript has been substantially improved by following your suggestions. We have provided point-by-point responses to the reviewer’s comments, as noted below. The revisions have been developed in consultation with all coauthors, and each author has approved the final version. However, it seems that comments 1 to 13, including "Main text 1” in which you wrote: “Lesion location was defined according to SYNTAX score. But actually …" do not correspond to this manuscript. It is considered that these may be responses to other papers that have accidentally been inserted. Please kindly verify this. 

Thank you for your assistance.

Reviewer 1

Comments to the Author

Abstract: 1. 

It is unclear from the title, as well as from the abstract, for which vessels the device is developed. Is that a thrombectomy device for coronaries? For central nervous system? For pulmonary? For deep veins?

♦ Authors’ response

Thank you for commenting on this important point.

We have developed a thrombectomy device for the central nervous system. To be more specific, it is a device to clear the occlusion of an intra- or extra-cranial artery that caused a cerebral infarction.

The above is indicated in the title and abstract.

♦ Changes 

(Page 1, Line numbers 1-2) Title

“In vivo evaluation of histopathologic findings of vascular damage after mechanical thrombectomy with the Tromba device in a canine model of cerebral infarction.”

(Page 2, Line numbers 22)

“A novel stent retriever device for in vivo mechanical thrombectomy for acute cerebral infarction has been developed.”

(Page 2, Line numbers 27)

“Through this experimental evaluation, we aimed to assess the safety and efficacy of the newly developed thrombus removal device for cerebral infarction.”

Abstract: 2. 

It has to be declared in the abstract, at what timepoint the animal models were sacrificed and analyzed. Considering that endothelial damage has the potential for instantaneous risk of further thrombus formation, therefore acute result is a critical aspect. On the other and it might heal over time, losing its relevance. But this would be nice to demonstrate. Since authors mention ‘intimal proliferation’, one would assume that chronic results are presented, however it is still unclear.

♦ Authors’ response

We agree with the reviewer’s comment. To rectify this, we have added the requested information (the time point of sacrifice and analysis) in the abstract.

♦ Changes 

(Page 2, Line numbers 31-33)

Angiographic and histopathologic evaluations were performed 1 month after the blood vessels underwent mechanical thrombectomy.

Abstract: 3. 

‘Moderate-’ and ‘slight narrowing’ are quite subjective phrasing for someone who is not fully familiar with the Grandin scale and therefore difficult to interpret for comparison. Please quantify it.

♦ Authors’ response

We appreciate your comment. The grade of reperfusion patency was defined according to the scale introduced by Grandin et al., as follows: no narrowing, slight narrowing (25% reduction in the lumen diameter), moderate narrowing (25%–50% stenosis or 50%–75% stenosis affecting only a short segment of the vessel), and severe narrowing (50%–75% stenosis affecting a long segment of the vessel or stenosis > 75%). We have added the abovementioned quantified information in the grade of reperfusion patency of the abstract.

♦ Changes 

(Page 2, Line numbers 34-40)

In the experimental group, the reperfusion patency was classified as “no narrowing” in five cases and “moderate narrowing (25%–50% stenosis)” in one case. In the control group, the reperfusion patency was classified as “no narrowing” in four cases, “moderate narrowing (25%–50% stenosis)” in one case, and “slight narrowing (less than 25% stenosis)” in one case.

Main text:1. 

It would be important to define, why the Solitaire FR was selected as reference. Is that the most widely used and accepted device of the field? 

♦ Authors’ response

The Solitaire FR Revascularization Device (ev3, Irvine, California) is the first dedicated blood flow restoration and thrombectomy dedicated device for the treatment of acute stroke. [Mordasini P et al, Am J Neuroradiol. 2010;31(5):972-8]

A recent pivotal study investigated whether mechanical thrombectomy could accelerate recanalization and increase the rate of revascularization. In most studies, the Solitaire FR Revascularization Device was most commonly used and these studies demonstrated rapid blood flow recovery and improvement of functional outcome after acute ischemic stroke.

♦ Changes 

(Page 4, Line numbers 51-57)

“Several recent pivotal studies have investigated whether mechanical thrombectomy can accelerate recanalization and improve stroke prognosis using a self-expanding stent retriever such as the Solitaire FR Revascularization Device. These studies have demonstrated that mechanical thrombectomy leads to rapid blood flow recovery and improvement of functional outcome in acute ischemic stroke patients [1-6].”

Main text: 2. 

From main text it is clear that results were ‘only’ at intermediate term after 30-days assessed. Since acute injury could have been assessed by intravascular imaging modalities, I believe the lack of acute data is a major limitation and it has to be acknowledged. 

♦ Authors’ response

We fully agree with the reviewer's comments.

In this study, imaging follow-up was performed immediately after mechanical thrombectomy and 1 month after mechanical thrombectomy, without regular follow-up investigations in between. Therefore, angiographic evaluation was not performed during the condition of acute phase vascular damage and blood flow, which is a limitation. This was noted in the limitations section.

♦ Changes 

(Page 15, Line numbers 269-272)

“Third, imaging follow-up was performed immediately after mechanical thrombectomy and 1 month later, without regular follow-up investigations in between. Therefore, angiographic evaluation was not performed during the condition of acute phase vascular damage and blood flow.”

Main text: 3. 

In the histopathologic result section the Authors write about swine model instead of canine. Please clarify. 

♦ Authors’ response

I apologize for this typographical error. We have changed “swines” to "canines."

Main text: 4.

In the limitation section Authors state: “However, histologic comparisons involving a large number of animals have been conducted in other canine models, and the results were considered to be of sufficient value.” It is unclear, which study and what device they refer to. It should be at least referenced or even described in a bit more details. 

♦ Authors’ response

We intended to conduct a study on relatively more canine models than other studies that had performed histological verifications through animal experiments. For the studies we referenced, Gralla J et al. conducted a study with 8 swines, and Mordasini et al. conducted a study with 5 swines. Brooks et al. studied 6 canines, and Nogureira et al. performed a biopsy with 2 swines and one canine. The following journals are attached as references. [11, 14, 16, 21-23]

1. [11] Gralla J et al. A dedicated animal model for mechanical thrombectomy in acute stroke. Am J Neuroradiol. 2006;27(6):1357-61

2. [14] Gralla J et al. Mechanical thrombectomy for acute ischemic stroke - Thrombus-device interaction, efficiency, and complications in vivo. Stroke. 2006;37(12):3019-24

3. [16] Nogueira RG et al. The Trevo device: preclinical data of a novel stroke thrombectomy device in two different animal models of arterial thrombo-occlusive disease. J Neurointerv Surg. 2012;4(4):295-300

4. [21] Mordasini P et al. In Vivo Evaluation of the First Dedicated Combined Flow-Restoration and Mechanical Thrombectomy Device in a Swine Model of Acute Vessel Occlusion. Am J Neuroradiol. 2011;32(2):294-300

5. [22] Brooks OW et al. A canine model of mechanical thrombectomy in stroke. J Neurointerv Surg. 2019;11(12):1243-8

6. [23] Mordasini P et al. In Vivo Evaluation of the Phenox CRC Mechanical Thrombectomy Device in a Swine Model of Acute Vessel Occlusion. Am J Neuroradiol. 2010;31(5):972-8.

♦ Changes 

(Page 15, Line numbers 266-267)

However, histologic comparisons involving a large number of animals have been conducted in other studies using swine or canine models (usually less than 10 animals).[reference]

Main text: 5. 

A certain structural / mechanical / design comparison of the novel versus the traditional device, focusing on potential advantages, would be important for better understanding.

♦ Authors’ response

I agree with your comment. Structural, mechanical, and design advantages and disadvantages were compared by comparing the stent structure developed by us, "the hybrid structure with open and closed cells", with "the closed cell" structure of the traditional device including the Solitaire stent. The content has been inserted into the discussion section and Figure 1B has been added to help readers understand the above. 

♦ Changes 

(Page 14, Line numbers 246-257)

“The thrombus removal part of the existing thrombus retrievable stent developed so far, including Solitaire, is in the form of a closed cell structure in which all cells are connected by a link to fix and recover the thrombus.. The stent of the "closed cell" structure fixes the thrombus in a relatively stable manner, so that the captured thrombus is smoothly recovered. However, due to its low flexibility and expansion power, it is difficult for this stent to adapt to the tortuous cerebral blood vessels, making it difficult to remove a thrombus in a severely bent region. This is a limitation as repeated thrombus removals can damage blood vessel walls. The hybrid cell structure is judged to be ideal because it has both the characteristics of an open cell, i.e., it can adapt well to complex and tortuous cerebral blood vessels, and a closed cell, i.e., it can stably remove and recover blood clots. In addition, a structure with a weak radial force and a structure that cannot be recaptured are the biggest disadvantages of an open cell. These can be overcome with a hybrid cell structure.”

Reviewer #2

Solid data presentation with regard to this new technology

Comments:

-With the assumption that acute stroke should be acute emboli acute thrombotic clots were injected; seems reasonable. But in reality also chronic clots could migrate to the cerebral arteries, eg patient has AF. Are there any data if this technology also works in chronic clots??

♦ Authors’ response

I understand the reviewer's question. A recent pivotal study investigated whether a mechanical thrombectomy could accelerate the potency and increase the rate of recanalization. The 5 trials are MR CLEAN, EXTEND-IA, ESCAPE, REVASCAT, and SWIFT PRIME, all published in the "New England Journal of Medicine". All these trials were performed in a similar way, and the group that underwent mechanical thrombectomy using a retrievable stent was compared with the group that did not. In most studies, mechanical thrombectomy with a retrievable stent showed rapid blood flow recovery and improved functional outcome in acute ischemic stroke. Furthermore, in most of the studies, a large number of participating patients had cardiac embolic infarction (including AF) leading to vascular occlusion due to chronic clots. 

In MR CLEAN, atrial fibrillation accounted for 28.3% and 25.8% of patients in the intervention and control groups, respectively, in EXTEND-I, 31% and 34%; ESCAPE, 37% and 40%; REVASCAT, 34% and 35.9%; and SWIFT PRIME, 39% and 36%, respectively. More than 30% of these patients had cardiac embolic infarction associated with AF, which can be represented by chronic clots, demonstrating that mechanical thrombectomy is effective in thrombus removal in this patient group as well. Therefore, data relating to this technique shows that it will be effective in chronic blood clots.

 The referenced papers are listed below.

-Reference-

1. Berkhemer OA, et al. A randomized trial of intraarterial treatment for acute ischemic stroke. New Engl J Med. 2015;372(1)

2. Campbell BCV, et al. Endovascular Therapy for ischemic stroke with perfusion-imaging selection. New Engl J Med. 2015;372(11):1009-18

3. Goyal M, et al. Randomized assessment of rapid endovascular treatment of ischemic stroke. New Engl J Med. 2015;372(11):1019-30

4. Jovin TG, et al. Thrombectomy within 8 hours after symptom onset in ischemic stroke. New Engl J Med. 2015;372(24):2296-306

5. Saver JL, et al. Stent-retriever thrombectomy after intravenous t-PA vs. t-PA alone in stroke. New Engl J Med. 2015;372(24):2285-95

-The authors state throughout the manuscript that they used 12 canine models (beagles), but in the section histopathological findings they state that they used 12 swines?? This need to be clarified

♦ Authors’ response

I apologize for this typographical error. We have changed “swines” to "canines."

---

## [Decision Letter · Decision Letter 1]

20 Jul 2022

PONE-D-21-40394R1In vivo evaluation of histopathologic findings of vascular damage after mechanical thrombectomy with the Tromba device in a canine model of cerebral infarctionPLOS ONE

Dear Dr. LEE,

Thank you for submitting your manuscript to PLOS ONE. After careful consideration, we feel that it has merit but does not fully meet PLOS ONE’s publication criteria as it currently stands.  Please can you address the following editorial requests in a revised manuscript: 1) In line with the PLOS data policy, please provide the raw histopathology images as supporting information files. Please also confirm in your response to reviewers document that all content in the supporting information files may be published under a CC-BY license, i.e. that there are no confidentiality restrictions. 2) Outline the rationale for using a canine model in your introduction or methods section.  3) Include the following information in the methods section of the main text of the manuscript (some of this is currently detailed in the SI files, but it should be moved to the main text):- The source of the animals.- Details of housing, husbandry, and care, including diet, enrichment, and exercise as applicable.- Animal health monitoring, including frequency and criteria and any efforts made to reduce suffering and distress, such as administering analgesics.- Whether humane endpoints were in place during the study and how they were applied

- The methods of anesthesia and euthanasia.- Any mortality that occurred outside of planned euthanasia or humane endpoints. In particular, please include a description of the euthanasia and vessel retrieval before histopathology, and a description of the care of the beagles post-surgery after in effect having an induced stroke.

Kind regards,

Joseph Donlan

Editorial Office

PLOS ONE

Journal Requirements:

Reviewers' comments:

Reviewer's Responses to Questions

**Comments to the Author**

1. If the authors have adequately addressed your comments raised in a previous round of review and you feel that this manuscript is now acceptable for publication, you may indicate that here to bypass the “Comments to the Author” section, enter your conflict of interest statement in the “Confidential to Editor” section, and submit your "Accept" recommendation.

Reviewer #1: All comments have been addressed

2. Is the manuscript technically sound, and do the data support the conclusions?

Reviewer #1: Partly

3. Has the statistical analysis been performed appropriately and rigorously? 

Reviewer #1: Yes

4. Have the authors made all data underlying the findings in their manuscript fully available?

Reviewer #1: Yes

5. Is the manuscript presented in an intelligible fashion and written in standard English?

Reviewer #1: Yes

6. Review Comments to the Author

Reviewer #1: (No Response)

7. PLOS authors have the option to publish the peer review history of their article (what does this mean?). If published, this will include your full peer review and any attached files.

Reviewer #1: No

---

## [Author Response · Author response to Decision Letter 1]

11 Aug 2022

Thank you for your opportunity to revise our manuscript. We appreciate your careful review and constructive suggestions. We believe that the manuscript has been substantially improved following your suggestions. We have provided point-by-point responses to the reviewers’ comments below. The revisions have been performed in consultation with all co-authors, and each author has approved the final version.

Thank you for your assistance.

Reviewer Comments to the Author

1) In line with the PLOS data policy, please provide the raw histopathology images as supporting information files. Please also confirm in your response to reviewers document that all content in the supporting information files may be published under a CC-BY license, i.e. that there are no confidentiality restrictions.

♦ Authors’ response

Raw histopathological images were additionally submitted as supporting information files. We confirmed that all content in the supporting information files may be published under a CC-BY license.

2) Outline the rationale for using a canine model in your introduction or methods section.

♦ Authors’ response

We have added the requested information to the Methods section.

♦ Changes 

(Revised Manuscript with Track Changes - Page 5, Line numbers 94-98) 

Canine models have several advantages. The size and extent of vasospasm occurring in canines are more similar to those in humans than in porcine and rodent models, allowing for more targeted endovascular treatment and the branches of the external carotid artery in the canine model have a vessel size similar to that of the human middle cerebral artery, which is sufficient for the study of novel endovascular devices for human cerebral blood vessels [11].

[11]. Mehra M, Henninger N, Hirsch JA, Chueh J, Wakhloo AK, Gounis MJ. Preclinical acute ischemic stroke modeling. Journal of Neurointerventional Surgery 2012;4:307-313.

3) Include the following information in the methods section of the main text of the manuscript (some of this is currently detailed in the SI files, but it should be moved to the main text):

- The source of the animals.

- Details of housing, husbandry, and care, including diet, enrichment, and exercise as applicable.

- Animal health monitoring, including frequency and criteria and any efforts made to reduce suffering and distress, such as administering analgesics.

- Whether humane endpoints were in place during the study and how they were applied

- The methods of anesthesia and euthanasia.

- Any mortality that occurred outside of planned euthanasia or humane endpoints.

In particular, please include a description of the euthanasia and vessel retrieval before histopathology, and a description of the care of the beagles post-surgery after in effect having an induced stroke.

 ♦ Authors’ response

We fully agree with the reviewer’s comment. The content has been inserted into the “Animal care and process of in vivo study” section of the Methods.

♦ Changes 

(Revised Manuscript with Track Changes - Page 5, Line numbers 90-132)

 To identify all experimental environments, we tested canines of the same age, weight, and size. Twelve 6-month-old canines (beagle dogs obtained from Marshall BioResources, North Rose, NY, USA) weighing between 6.0 and 8.0 kg were included in this study. Canine models have several advantages. The size and extent of vasospasm occurring in canines are more similar to those in humans than in porcine and rodent models, allowing for more targeted endovascular treatment and the branches of the external carotid artery in the canine model have a vessel size similar to that of the human middle cerebral artery, which is sufficient for the study of novel endovascular devices for human cerebral blood vessels [11]. One animal was bred in a cage (0.9 m2) in accordance with AAALAC guidelines (0.74 m2), and an individual identification card with the test and animal numbers was attached to the breeding cage. A classic dog toy was used for environmental enrichment. The conditions of the breeding environment were as follows: temperature, 23°C ± 3°C; relative humidity, 55% ± 15%; ventilation, 10–20 times/h; lighting time, 12 h (lights up at 8 am, lights out at 8 pm), and illuminance, 150–300 Lux. This study was conducted in an established canine-breeding room. For the feed supplied to the test, a solid feed for test animals (2925 Telan Global 25% Protein Dog Diet) was sterilized by irradiation. Ultrapure water produced by an RO water (reverse osmosis distilled water) production and supply device was sterilized using sodium hypochlorite and a UV device and then freely dispensed through an automatic water supply nozzle.

 All experiments were conducted under general anesthesia. Preoperative anesthesia was induced by intramuscular injection of 5 mg/kg Zoletil (Bibac Korea, Seoul, Korea) and 2 mg/kg Rompun (Bayer Korea, Ansan, Korea). For postoperative pain/stress relief in experimental animals, 0.2 mg/kg of Metacam (Boehringer Ingelheim, Germany) was subcutaneously injected once daily for three days. In addition, 15 mg/kg cefazolin (Chong Kun Dang, Korea) was subcutaneously injected once daily for three days to prevent infection. For humane management of diet, exercise, and nutrition in experimental animals with stroke induced by vascular occlusion and mechanical thrombectomy, the experiment was limited to unilateral sites of the carotid artery, not to bilateral sites. During the observation period, animals treated with the stent retriever in the breeding room of the experimental animal center were visually observed for abnormal symptoms once a day, and the experimental animals had no restrictions on their daily activities, including diet.

 Histopathological analysis was performed by excising vessels that had undergone mechanical thrombectomy. Anesthesia was induced by intramuscular injection of 5 mg/kg Zoletil (Bibac Korea, Seoul, Korea) and 2 mg/kg Rompun (Bayer Korea, Ansan, Korea) in the operating room of the experimental animal center. After induction of general anesthesia, airway intubation was performed, the animal was connected to an inhalation anesthesia machine (Fabius GS premium, Drager, Germany), and respiratory anesthesia was maintained with isoflurane (<3%). The animals’ anesthetic conditions (electrocardiogram, oxygen saturation, carbon dioxide concentration, etc.) were constantly monitored using a patient monitoring system (Vista 120, Drager, Germany). At the end of the experiment, 40 ml of KCl was injected intravenously after intramuscular injection to induce euthanasia, and respiration and cardiac arrest in the experimental animals were confirmed. There were no other mortalities, except planned euthanasia, and no humane endpoints in the study.

---

## [Editor Report · Decision Letter 2]

29 Sep 2022

In vivo evaluation of histopathologic findings of vascular damage after mechanical thrombectomy with the Tromba device in a canine model of cerebral infarction

PONE-D-21-40394R2

Dear Dr. LEE,

We’re pleased to inform you that your manuscript has been judged scientifically suitable for publication and will be formally accepted for publication once it meets all outstanding technical requirements.

Kind regards,

James Mockridge

Staff Editor

PLOS ONE

---

## [Editor Report · Acceptance letter]

6 Oct 2022

PONE-D-21-40394R2 

In vivo evaluation of histopathologic findings of vascular damage after mechanical thrombectomy with the Tromba device in a canine model of cerebral infarction 

Dear Dr. Lee:

I'm pleased to inform you that your manuscript has been deemed suitable for publication in PLOS ONE. Congratulations! Your manuscript is now with our production department. 

Kind regards, 

on behalf of

Dr James Mockridge 

Staff Editor

PLOS ONE